# Measuring reading time: Comparing logged and self-reported data in relation to reading skills

**Brice Brossette**[1,2]*, **Laurie Persia-Leibnitz**[1,3], **Mee-Jin Chalbos**[1],
**Chloé Prugnières**[1,2], **Stéphanie Ducrot**[1,2,4]

**1** Aix-Marseille Univ, CNRS, LPL, Aix-en-Provence, France, **2** Aix-Marseille Univ, Pôle Pilote AMPIRIC, Marseille, France, **3** Academy of Martinique, Les Hauts de Terreville, Schoelcher, France, **4** Institute for Language, Communication, and the Brain, Aix-Marseille Université, Aix-en-Provence, France

* brice.brossette@gmail.com

## Abstract

### Background

Children's reading time at home plays a critical role in their reading development. However, existing measures of reading time, based on self-reports, are often biased. Logged data from mobile apps may offer a more reliable alternative, as shown in studies examining screen time in digital media use.

### Objectives

This study compared logged and self-reported measures of reading time and examined their associations with reading skills in French primary school children.

### Method

One hundred and nine children from Grade 1 to Grade 5 and their parents participated. Parents completed a retrospective questionnaire estimating weekly reading time (self-reported measure). They then used a mobile application to record their child's reading activities in real time over a 14-day period (logged measure). All children were assessed on their reading fluency.

### Results and conclusions

The self-reported measure yielded significantly higher reading time estimates (M = 6.26 hours/week) than the logged measure (M = 2.11 hours/week), with a moderate correlation between the two (r = .45). Crucially, the logged measure showed stronger predictive validity for reading fluency (r = .39) than the self-reported measure (r = .25). Regression analyses confirmed that when both measures were included simultaneously, only the logged reading time remained a significant predictor of reading performance. These findings suggest that logged measures obtained via ambulatory assessment (here, using a mobile app) provide more accurate estimates

**Data availability statement:** All data and code are available at the Open Science Framework (https://osf.io/5ky7w/).

**Funding:** This research was supported by a postdoctoral fellowship awarded to B.B. by the Pilot Center for Research in Education and Teacher Training (AMPIRIC, France 2030, "Territories of Educational Innovation" initiative, operated by the Caisse des Dépôts), and by funding from the Carnot Cognition Institute awarded to S.D. (EXPOLECT_DOM).

**Competing interests:** The authors have declared that no competing interests exist.

of reading time and superior predictive validity compared to traditional self-reports. This methodology offers promising avenues for future research on reading habits and literacy development.

---

## 1. Introduction

The development of reading competencies is a central priority in educational public policies, which promote a broadened conception of literacy that goes beyond basic decoding skills and emphasizes a functional understanding of written language [1,2]. Developing strong reading competencies requires well-established reading comprehension skills [3]. These skills are regularly assessed in large-scale international studies [4], which consistently show that children's learning environments play a central role in their ability to comprehend written materials [5]. In particular, print exposure appears to be an important explanatory factor [6].

The mechanisms underlying this effect have been well described within the Home Literacy Model [7,8]. This model posits that both formal and informal home literacy activities support early literacy development, as well as oral language and vocabulary growth. Moreover, early reading routines have been shown to predict later engagement in leisure reading [9], which in turn contributes to the development of reading skills [10], while the reciprocal relationship also holds [11]. This reciprocal relationship underlies the "snowball effect," a positive feedback loop in which reading becomes increasingly enjoyable, thereby fostering skill development, confidence, and motivation to continue reading, all of which are crucial for the development of reading self-efficacy over time [12–14].

Importantly, this reciprocal relationship is developmentally dynamic rather than linear. Longitudinal and genetically informed studies indicate that during the early school years, reading ability primarily predicts later engagement with print [11,15]. From adolescence onward, the direction of this relationship tends to reverse, such that greater exposure to written language plays a stronger role in reading development [16,17]. However, despite this body of evidence, the directionality of effects between print exposure and reading skills remains an open and debated question, with no clear consensus in the literature. Importantly, even when reading skills are considered primary, promoting print exposure may act as a protective factor, particularly for children with reading difficulties or those growing up in disadvantaged environments [18]. This appears particularly important because schools do not always succeed in compensating for a deprived family environment that does not provide sufficient reading time [19].

Over the past decades, research has employed diverse measures to estimate reading time [20]. Nonetheless, reliance on self-reported data may partly explain the heterogeneity and contradictory findings observed in the literature regarding the causal direction between reading time and reading skills. In the following section, we discuss the limitations associated with such self-reported measures. We then explore an innovative approach to assessing reading time through ambulatory assessment,

inspired by digital log-based methods commonly used to assess screen time [21]. We believe that these innovative measures will allow for a more accurate estimation of reading time and its effects on the development of reading skills.

## 1.1 Traditional self-reported measures of reading time

Traditionally, reading time has been assessed using retrospective self-report questionnaires [20]. These measures show small to moderate associations with reading outcomes [12] and capture both quantitative aspects (e.g., time and frequency) and qualitative ones (e.g., text type, parent-child interaction), which differentially impact reading motivation and skill development [22–25]. However, retrospective questionnaires require complex cognitive judgments and are therefore vulnerable to multiple source of bias, including subjective interpretations of frequency and duration, memory and reference biases, and social desirability effects, often leading to overestimation of actual reading behavior [26–33].

As retrospective self-reported measures appear too coarse-grained, some researchers have turned to more fine-grained approaches, such as reading diaries. These involve daily reporting of reading activities, capturing not only the time spent reading but also qualitative aspects, such as the type of material read [34–36]. Such diaries have been found to be reliable predictors of reading outcomes and motivation [35,37–39]. Although they reduce some limitations of retrospective questionnaires, their data quality depends on compliance and timing of completion, as delayed entries may reintroduce retrospective bias and increase missing data, particularly over longer collection periods [40–42]. In addition, traditional paper-and-pencil formats impose a substantial burden on participants, which may negatively affect adherence and data quality [43].

## 1.2 Ambulatory assessment of reading time

To address these limitations, some researchers have proposed using ambulatory assessment [44,45], defined as the use of digital tools to track behaviors in real time and in natural settings. Thanks to technological advances, ambulatory assessments can now be easily carried out using mobile apps on participants' phones, as already demonstrated in the field of health [46,47]. More specifically, these methods have recently been successfully applied to assess screen time, demonstrating stronger psychometric properties than self-reported measures (for a review, see [48]).

However, to date, only one study has used ambulatory assessment to measure reading time, despite several advantages highlighted by Locher and collaborators [49]. First, smartphone ownership is common, even in disadvantaged families, enabling continuous data collection through a dedicated mobile app. This provides a convenient way to complete a reading diary at any time. Moreover, mobile apps offer high flexibility by allowing questions to be tailored, either by removing unnecessary items or by adding new ones based on participants' responses or profiles. Another advantage is the ability to detect when entries are completed retrospectively, allowing for a more precise evaluation of data quality. Finally, mobile apps may be easier to use than paper-based methods, promoting broader participation and increasing the generalizability of results. Locher and collaborators [49] demonstrated that using a mobile app to track the reading behavior of university students is a reliable approach that reduces the burden of diary completion. Importantly, they found that app-based data and retrospective questionnaires were closely related, although reading time was overestimated in the questionnaire. This suggests that mobile apps may help reduce social desirability bias, which contributes to such overestimations.

## 1.3 The present study

The present study aimed to extend this approach in several ways. First, we developed a mobile application that allowed participants to time the duration of reading activities using a stopwatch function, rather than reporting it retrospectively. We believed that this method would encourage greater attention to accurately reporting reading activity, while reducing retrospective bias and providing a more objective measure of reading time. To distinguish this approach from traditional

self-reports, and to draw a parallel with studies using logged screen time in digital media research, we refer to this method as a *logged measure of reading time*. Second, the study focused on primary school children. Since children do not necessarily own smartphones, we targeted reading time at home, where parents could use the mobile app to record their child's reading activities. Finally, we assessed children's reading fluency to examine its relationship with recorded reading times. A total of 109 children from Grade 1 to Grade 5 and their parents first completed a paper-and-pencil questionnaire about reading behavior, then used a mobile app for 14 days to log the duration of reading activities in real time. Children's reading fluency was assessed at school. The study aimed, first, to evaluate the degree of convergence between self-reported and app-recorded reading time, and second, to assess the extent to which each measure contributes to the prediction of reading fluency.

## 2. Method

### 2.1. Participants

A total of 109 children (49 female, 60 male) from a public elementary school (Grade 1: $n = 20$; Grade 2: $n = 29$; Grade 3: $n = 29$; Grade 4: $n = 13$; Grade 5: $n = 18$) were tested in the middle of their school year (from 01/01/2024 to 30/04/2024). All participants were native French speakers with normal or corrected-to-normal vision. Caregivers provided written informed consent prior to the experiment. In addition, children provided verbal assent before participation. The experimenter explained the study in age-appropriate language, emphasized that participation was voluntary, and informed children that they could withdraw at any time without consequences. This procedure, including the use of verbal child assent, was reviewed and approved by the French Ethics Committee Review Board (2023-01-05-04). The experiment was conducted in accordance with relevant guidelines and regulations, as well as the Declaration of Helsinki.

### 2.2. Measures of reading time

#### 2.2.1. Self-report measure: The paper-and-pencil questionnaire.
A self-report measure of children's reading time was obtained using a paper-and-pencil questionnaire completed by parents (similar to that used in [50]). The questionnaire asked parents to retrospectively estimate the average number of hours per week their child spent engaged in three distinct types of reading activities at home: (1) school-related reading, defined as the time parents supervised their child while completing reading assignments for school; (2) shared reading, referring to time spent reading to or with the child outside of homework; and (3) independent reading for pleasure, referring to time the child spent reading alone by choice. These distinctions are important, as the nature of children's reading activities is known to evolve with reading development: shared reading tends to decrease as decoding becomes more automated, while independent reading increases [51,52]. By evaluating all three components, the questionnaire provides a comprehensive picture of the overall volume of exposure to written material across the elementary school years. For the purposes of the present study, weekly time estimates for the three activity types were summed to create a single composite index of reading time (see Results section). This questionnaire-based approach constitutes a self-report measure of reading time, as it relies on parental estimates rather than direct behavioral tracking.

#### 2.2.2. Logged measure: The reading diary app data.
A logged measure of children's reading time was obtained using a custom-built mobile application developed with FlutterFlow [53]. Parents were asked to use the app to time their child's reading activities over a 14-day period. This period was selected based on previous work ([49] informed by [40]). At the end of each recorded session, parents were prompted to indicate the type of reading activity that had taken place: (1) school-related reading, (2) shared reading, or (3) independent reading for pleasure—categories that were strictly identical to those used in the paper-and-pencil questionnaire. The total duration of each type of activity was automatically logged by the app. To produce a measure of average weekly reading time comparable to the questionnaire data, the total duration across the 14-day period was divided by two

(see Results section). This approach provides a time-based, behaviorally grounded measure of children's reading time in the home environment. Unlike retrospective estimates, it offers a chronometric index of real-world reading behavior.

### 2.3. Measure of reading fluency

To assess reading fluency, we used the Alouette Test [54], a standardized reading assessment widely used in France for children aged 5–14. In this task, participants are asked to read aloud a 265-word text within a time limit of three minutes (180 seconds), aiming for both speed and accuracy. The text is designed to minimize reliance on semantic cues by presenting syntactically correct but nonsensical sentences (e.g., "Le printemps a mis ses nids," [Spring has put on its nests]), thereby targeting decoding skills. It contains rare vocabulary, irregular spellings, and orthographic challenges such as silent letters and phonologically confusable items. Three main variables were recorded: the number of correctly read words (M), total reading time in seconds (TL), and the number of reading errors (E). These variables were used to compute the reading fluency index (CTL = [(M-E))/TL * 180]).

## 3. Results

### 3.1. Preprocessing of reading time data

**3.1.1. Self-reported reading time.** Following data collection, a small number of missing values were identified in the parent-reported duration estimates ($n = 5$). Specifically, two values were missing for school-related reading activities, one for shared reading, and two for independent reading for pleasure. To address these missing entries, we imputed the corresponding values using the mean duration reported by other children in the same grade level for the same type of activity. In addition, to minimize the influence of outliers, a winsorization procedure was applied to the upper 5% of values for each activity type (for more details, see [55]). Extreme values were replaced with the highest non-extreme value within the same distribution, thereby reducing the weight of atypical responses while preserving rank order. For each participant, weekly durations across the three activity types were then summed to generate a single composite index of total reading time

**3.1.2. Logged reading time.** A total of 1,324 reading activities were recorded via the mobile application across 109 participants. Reading activities shorter than one minute were excluded as unlikely to reflect meaningful reading behavior (e.g., premature stops or test entries). These accounted for 1.36% of the total data. This decision was based on feedback from parents indicating occasional handling errors. Because parents were not allowed to delete recorded activities in order to preserve data quality, the corresponding corrections were performed manually after data collection. Following this exclusion, two participants had no remaining valid reading entries and were therefore removed from further analyses. Extremely long reading sessions, due to a failure to stop the timer, were replaced with the participant's mean reading time (3.68% of the data).

Measurement quality was assessed using two complementary approaches. First, internal consistency was estimated following the procedure outlined by Locher and collaborators [49], treating daily reading time as repeated measurements of the same latent construct—namely, time spent reading. The resulting Cronbach's alpha was .67, approaching the conventional threshold for acceptable reliability. Second, test–retest stability was evaluated by correlating total reading time between week 1 and week 2. The correlation was moderate and statistically significant *(r=.48, p<.001)*, providing further support for the temporal stability of the app-based measure.

### 3.2. Comparisons between self-reported and logged measures of reading time

A comparison of reading time estimates revealed substantially higher weekly durations for the self-reported measure (M = 6.26 hours, SD = 2.78) than for the logged measure (M = 2.11 hours, SD = 1.80). This difference was highly significant

($t = 12.97$, $p < .001$, $d = 1.64$). Reading time did not significantly vary across grade levels (see Table 1; all pairwise comparisons with p > .05).

Despite large differences in absolute levels, the two measures were moderately correlated ($r = .45$, $p < .001$), suggesting that the two measures capture partly overlapping constructs. To further evaluate the consistency of individual rankings across the two measurement methods, we computed an intraclass correlation coefficient (ICC) using a mixed-effects model implemented via the R package *psych* [56]. The ICC indicated a significant but relatively low level of agreement between the two measures ($F (106, 106) = 2.42$, $p < .01$).

To further assess the agreement between the two measurement methods, we conducted a Bland–Altman analysis [57]. The 95% limits of agreement (grey dotted lines) ranged from approximately 0 to −8 hours, indicating wide individual differences and poor agreement, consistent with the ICC results. Fig 1 shows that self-reported reading time was over-estimated by approximately 4 hours per week (grey dashed line). A systematic bias was observed for most participants (grey points), with the self-reported measure consistently reporting higher values. This bias appeared proportional, as the variability in differences increased with higher mean reading durations. A linear regression confirmed this trend, showing a significant proportional bias (grey solid line).

### 3.3. Relation between reading time and reading fluency

To examine the relationship between reading time and reading fluency, we first conducted Pearson correlation analyses. Weekly logged reading time recorded via the mobile application showed a moderate and significant correlation with fluency scores ($r = .39$, $p < .001$). Self-reported reading time from the paper-based questionnaire was also positively correlated with fluency, although to a lesser extent ($r = .25$, $p < .05$).

We then ran a series of linear regression models to compare the predictive power of the two measurement methods, controlling for age, gender, and mother's professional occupation. In the self-reported model, weekly reading time significantly predicted fluency performance (b = 0.09, $SE = 0.04$, $\beta = 0.23$, *partial* $R^2 = .06$, $p < .05$). Mother's occupation also emerged as a significant predictor (b = −0.21, $SE = 0.08$, $\beta = −0.23$, *partial* $R^2 = .06$, $p < .05$). This model accounted for 9.2% of the variance in fluency scores (adjusted $R^2 = .092$).

In contrast, the logged model showed that objectively measured weekly reading time was a stronger and more reliable predictor of fluency (b = 0.23, $SE = 0.06$, $\beta = 0.37$, *partial* $R^2 = .13$, $p < .001$), while mother's occupation was not significant (b = −0.15, $SE = 0.08$, $\beta = −0.17$, *partial* $R^2 = .03$, $p = .070$). The model explained 16.3% of the variance in reading fluency (adjusted $R^2 = .163$).

Table 1. **Mean weekly reading time (in hours) by grade level, measured using a mobile application (logged measure) and a paper-pencil questionnaire (self-reported measure). For each grade, the table presents the mean and standard deviation (in parentheses), as well as the observed range [minimum; maximum].**

| Grade | Logged Measure (mobile-app) | Self-reported Measure (paper-pencil questionnaire) |
|---|---|---|
| 1 | 1.20 (.92) [.11; 3.22] | 5.93 (2.51) [2.00; 10.00] |
| 2 | 2.43 (1.62) [.32; 6.08] | 6.96 (3.36) [2.00; 15.80] |
| 3 | 1.94 (1.94) [.12; 8.37] | 6.24 (2.63) [1.50; 11.00] |
| 4 | 2.18 (1.66) [.31; 6.04] | 6.06 (2.77) [2.00; 11.00] |
| 5 | 2.82 (2.29) [.58; 7.13] | 5.69 (2.32) [2.00; 10.80] |

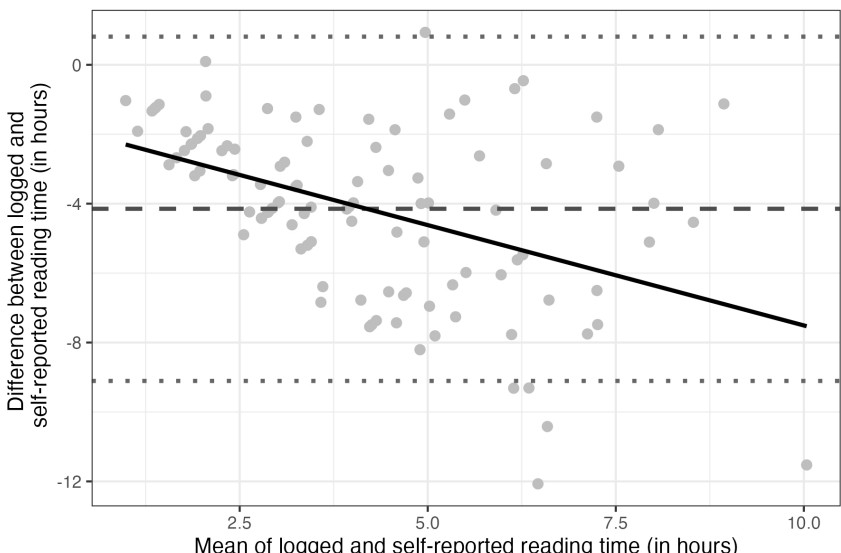

**Fig 1. Bland-Altman plot illustrating the agreement between logged and self-reported reading time (in hours).** The x-axis represents the mean of the two measures for each participant, and the y-axis shows the difference between logged and self-reported reading times. The solid black line represents the linear regression of the differences on the means, indicating a systematic bias. The dashed line indicates the average difference (bias), and the dotted lines indicate the limits of agreement.

Finally, in a combined model including both reading time measures, only the logged reading time remained a significant predictor of fluency ($b = 0.21$, $SE = 0.07$, $\beta = 0.33$, partial $R^2 = .08$, $p < .01$), whereas the self-reported estimate no longer reached significance ($b = 0.03$, $SE = 0.04$, $\beta = 0.08$, partial $R^2 = .01$, $p = .46$). Mother's occupation again failed to reach significance ($b = -0.16$, $SE = 0.08$, $\beta = -0.17$, partial $R^2 = .03$, $p = .066$). The model explained 15.9% of the variance in reading fluency (adjusted $R^2 = .159$).

## 4. Discussion

The present study evaluated the relevance of ambulatory assessment of reading time in primary school children using a smartphone-based app. The first objective was to examine the convergence between the logged measure and a traditional self-reported paper-and-pencil retrospective questionnaire. Our results showed that reading time was overestimated using the self-reported measure, although both measures were significantly correlated. However, their agreement remained low, indicating that participants who reported reading more on the questionnaire did not necessarily log more reading time in the mobile app.

The second objective was to assess the extent to which each measure predicts reading fluency. Results showed stronger associations between reading fluency and the logged measure. Regression analyses confirmed this pattern: although both measures were significant predictors when tested separately, only the logged measure remained significant when both were included in the same model. This suggests that the logged measure is more reliable and informative.

In the following discussion, we examine the data quality of both reading time measures and argue for broader use of log-based approaches in research. We then discuss how the type of measure affects the prediction of reading fluency scores. Finally, we discuss the implications and challenges of using log-based data in research on print exposure.

### 4.1. Data quality of the log-based measure

The log-based measure showed satisfactory data quality, with acceptable internal consistency and reporting stability over the two-week period. However, data quality appeared lower than that reported by Locher and collaborators [49] with university students. Several explanations can be considered.

First, logging relies heavily on parental compliance. Parents must consistently initiate and terminate the stopwatch for each reading episode, which may result in omissions or inaccuracies, particularly during busy routines or over extended monitoring periods. Second, independent reading episodes occurring outside parental supervision may go unrecorded. This limitation is likely to increase with children's age and autonomy, potentially leading to a systematic underestimation of reading time in older children. Third, the use of a stopwatch, as opposed to manual time entry, may introduce additional noise—for example when the timer is started unintentionally or not stopped immediately after a reading episode. Despite these limitations, the logged reading times remained very similar to those reported by Locher and collaborators [49] with university students. Importantly, these limitations indicate that app-based logged measure should not be interpreted as exhaustive records of all reading activity. Rather, they should be considered conservative, behaviorally anchored estimates of supervised reading time.

### 4.2. Data quality of self-reported measure

Contrary to the log-based measure, self-reported measure showed clear signs of overestimation. This overestimation may be due to social desirability bias, which could have been stronger in this study because total reading time included self-reports for three types of reading activities (shared reading, independent reading, and reading for homework), potentially amplifying the effect. In this regard, research on reading time rarely includes controls for social desirability bias [58], even though several methods have been developed to address it [59]. In contrast, the logged measure seems to be less sensitive to social desirability bias, as it is more difficult or impractical to inflate reading time when using a stopwatch.

Another, non-exclusive explanation for the overestimation in self-reports is that parents may have difficulty clearly distinguishing between different types of reading activities. As a result, a single reading event may be reported under multiple categories (e.g., both shared reading and reading for homework), leading to double counting and an inflated total reading time. The rationale for separating reading activities was to capture the diversity of reading practices across the primary school years. For example, Grade 1 children are more likely to engage in shared reading and less in independent reading, while the opposite tends to be true in Grade 5 [51,52]. Using a single global question that explicitly includes all types of reading activities might have reduced overestimation. In the same vein, it is possible that parents overestimate reading time because children appear to be reading (e.g., holding a book) without sustained engagement in decoding or comprehension. This limitation should be more pronounced for self-reports than for log-based measures. Stopwatch-based logging requires an explicit initiation of reading episodes, thereby reducing reliance on inference or routine-based assumptions. However, the present design did not allow for direct verification of children's cognitive engagement during reading episodes. Neither the questionnaire nor the app-based logging can distinguish between active reading and more superficial engagement.

### 4.3. What do these two measures really capture?

At first glance, the moderate correlation between the two measures suggests that logged and self-reported data capture only partially overlapping constructs, in line with findings from digital media use research, where similar discrepancies between logged and self-reported measures have been observed [48]. This is further supported by the low level of agreement between them, indicating that the measures are not interchangeable for consistently ranking participants. The Bland–Altman analysis revealed a systematic bias, with the self-reported measure overestimating reading time for nearly all participants by an average of four hours. This finding supports the idea that self-reports are sensitive to social

desirability bias, as previously discussed. However, this explanation is not entirely sufficient, as the bias was also proportional to the total reading time reported by participants. One possible explanation is that frequent readers tend to overreport their reading time, as it may be more difficult to estimate the duration of habitual activities embedded in daily routines [26]. It is also possible that less skilled readers, due to the cognitive effort required and reduced enjoyment, perceive time as passing more slowly, which may lead to overestimation in self-reported reading time [60].

The proportional nature of this systematic bias remains an open question, as other research has identified different patterns. In the field of digital media use, a regression-to-the-mean bias—where heavy users tend to underreport their activity, while light users tend to overreport it—has been described in several studies [61–63]. As acknowledged by Scharkow [63], several factors may influence the direction and nature of systematic bias, including gender, cognitive abilities, social desirability, behavior frequency, the specificity of the construct measured, and methodological limitations of the instruments. Systematic bias can distort not only correlations with other variables but also estimates of means, variances, and proportions. Future studies using logged data to measure reading time should aim to disentangle the contribution of these factors to better understand the biases affecting self-reported data, which appear less suitable for accurately capturing actual reading time.

#### 4.4. How are these two measures associated with reading fluency?

Both measures were significantly correlated with reading fluency, with the logged measure showing the strongest association. Regarding this difference, it is possible that mean imputation may have slightly attenuated correlations involving the self-reported measure. However, given the very small proportion of missing data and the use of grade-level imputation, this effect is likely negligible and unlikely to explain the observed superiority of the logged measure. Importantly, available evidence indicates that the present sample size was adequate to investigate associations between different measures of reading time and reading fluency. Based on prior literature, Torppa et al. (2020) reported correlations between leisure reading and reading fluency ranging from $r = .28$ to $.41$ ($M \approx .35$) in children from Grade 1 to Grade 4 [16]. These findings converge with the meta-analysis by Mol and Bus (2011) [12], which showed that associations with reading fluency are generally stronger than those observed for more constrained basic decoding skills. Adopting a conservative expected correlation of $r = .28$ and a conventional target power of 80% [64], an a priori power analysis using the pwr.r.test function from the R package pwr indicated that approximately 97 participants would be required (two-sided $\alpha = .05$). With the present sample size (N = 109), the corresponding post hoc power based on the observed correlation ($r = .25$) is approximately 75%, and the sample size required to detect this effect with 80% power would be approximately 123 participants.

This predictive advantage was further confirmed in the regression analyses: when both measures were included in the model, only the logged measure remained a significant predictor. This suggests that the self-reported measure no longer provided meaningful information, likely due to shared variance with the logged measure. This finding is consistent with the literature, which shows that daily measures of reading time are often considered the gold standard for assessing print exposure. They limit retrospective bias and offer superior content, convergent, and criterion validity [35,39]. Their predictive power is comparable to that of print exposure checklists, which have been shown to outperform self-reported reading time [12].

Moreover, the fact that mother's occupation was a significant predictor only in the self-reported model, but not in the logged model, supports the idea that more precise behavioral tracking can reduce reliance on proxy variables. Although this issue has not, to our knowledge, been directly examined in the field of literacy, it has been addressed in other domains. For example, in the insurance industry, behavioral tracking—such as installing devices in cars to record real-time driving behaviors (e.g., duration, distance, speed)—allows for a more accurate assessment of risk than relying on proxies like income or occupation [65]. It is well established that proxy variables can introduce measurement bias and may fail to capture the underlying construct. Taken together, these findings help explain the added value of the logged measure over the traditional self-reported measure.

### 4.5 Implications and challenges for future research

Despite these promising results, several avenues for improving the reliability of app-based logged measures can be identified. First, data quality could be improved. Allowing retrospective entries may reduce missing data. However, it may also encourage participants to abandon real-time logging. This would reintroduce recall bias and create heterogeneous data types [58,59]. A better solution would be to include a post-study compliance questionnaire. This would allow researchers to assess how consistently families used the stopwatch function. An annotation feature could also be added. Users could then report recording errors, such as forgetting to stop the timer.

Second, as discussed earlier, parental involvement may have influenced data completeness. In the present study, parents were asked to record their child's reading activities. It is possible that the degree of parental supervision varied depending on the child's age and level of autonomy. It may also have differed according to socioeconomic background. Such variability could have affected the proportion of missing data and the overall consistency of logging. Future studies should therefore include specific measures assessing children's autonomy in both reading practices and recording behaviors. An open question remains whether child self-recording would provide more complete or more reliable data. The answer likely depends on developmental factors and the child's level of autonomy.

Third, engagement and scalability must be considered. In the present study, research assistants supported families. This support improved compliance but limits scalability. Families more familiar with digital tools may also have been more likely to participate. This raises potential selection bias. These issues could be mitigated by improving the interface and simplifying the user experience. Gamification features may also enhance engagement. However, researchers should monitor possible reactivity effects induced by such features.

Improved engagement could make longer data collection periods more feasible. In the present study, we selected a 14-day period based on prior diary research [40,49]. Because app-based logging reduces participant burden, extended recording periods may be achievable without compromising data quality. This would make the method suitable for longitudinal designs and allow researchers to examine learning dynamics in greater detail.

**App-based logging also opens broader perspectives for cross-cultural research.** Applications can be developed in multiple languages. This may facilitate participation from families who do not speak the language of schooling [19]. Such an approach is particularly relevant because literacy practices in the home language can transfer to school-based literacy skills [66,67].

Finally, this approach is not limited to home literacy research. App-based logging could be implemented in school contexts. For example, it could complement existing measures of Academic Learning Time [68]. Current methods rely on teacher self-reports or direct classroom observations. The former are efficient but prone to bias. The latter are resource intensive. Real-time logging by teachers may offer a useful compromise.

### 4.6. Conclusion

While our findings may appear to argue for abandoning questionnaires, we believe they instead open new avenues for improving these measures. Rather than opposing the two approaches, their combined use in future studies could help identify the various factors that contribute to systematic biases, as widely documented in the literature. This, in turn, would support the development of statistical techniques to adjust for these influences [48,69]. Strengthening the predictive validity of questionnaires through such corrections would preserve their value as practical and scalable tools. Ultimately, integrating real-time and self-reported measures holds strong potential for advancing research on reading practices—both in ecological and methodological terms.

### Acknowledgments

We would like to thank Pierre Blache, Inspector of National Education, as well as the teachers who made it possible to implement the project in their schools and classrooms. We are also deeply grateful to the children and their parents for their participation in the study. Finally, we wish to acknowledge Florian Amerigo and Marylou Garabedian for their valuable assistance in data collection as part of their Master's thesis work.

**Declaration of generative AI and AI-assisted technologies in the writing process:** During the preparation of this work, the authors used ChatGPT to proofread the manuscript. After using this tool/service, the authors reviewed and edited the content as needed and take full responsibility for the content of the publication.

## Author contributions

**Conceptualization:** Brice Brossette, Stéphanie Ducrot.

**Data curation:** Brice Brossette, Laurie Persia-Leibnitz, Mee-Jin Chalbos, Chloé Prugnières, Stéphanie Ducrot.

**Formal analysis:** Brice Brossette.

**Funding acquisition:** Brice Brossette, Stéphanie Ducrot.

**Investigation:** Brice Brossette.

**Methodology:** Brice Brossette, Stéphanie Ducrot.

**Project administration:** Brice Brossette, Laurie Persia-Leibnitz, Mee-Jin Chalbos, Chloé Prugnières, Stéphanie Ducrot.

**Supervision:** Brice Brossette, Stéphanie Ducrot.

**Validation:** Brice Brossette.

**Visualization:** Brice Brossette, Stéphanie Ducrot.

**Writing – original draft:** Brice Brossette.

**Writing – review & editing:** Brice Brossette, Stéphanie Ducrot.

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
