## [Decision Letter · Decision Letter 0]

29 Dec 2025

Dear Dr. Brossette,

Thank you for submitting your manuscript to PLOS ONE. After careful consideration, we feel that it has merit but does not fully meet PLOS ONE’s publication criteria as it currently stands. Therefore, we invite you to submit a revised version of the manuscript that addresses the points raised during the review process.

We look forward to receiving your revised manuscript.

Kind regards,

Elena del Pilar Jiménez-Pérez, Ph.D.

Academic Editor

PLOS One

Journal Requirements:

2. In the ethics statement in the Methods, you have specified that verbal consent was obtained. Please provide additional details regarding how this consent was documented and witnessed, and state whether this was approved by the IRB.

4. Please update your submission to use the PLOS LaTeX template. The template and more information on our requirements for LaTeX submissions can be found at http://journals.plos.org/plosone/s/latex.

5. Please amend your list of authors on the manuscript to ensure that each author is linked to an affiliation. Authors’ affiliations should reflect the institution where the work was done (if authors moved subsequently, you can also list the new affiliation stating “current affiliation:….” as necessary).

6. Please upload a new copy of Figure 1 as the detail is not clear. Please follow the link for more information: https://journals.plos.org/plosone/s/figures

Reviewers' comments:

Reviewer's Responses to Questions

**Comments to the Author**

1. Is the manuscript technically sound, and do the data support the conclusions?

Reviewer #1: Yes

Reviewer #2: Yes

2. Has the statistical analysis been performed appropriately and rigorously?

Reviewer #1: Yes

Reviewer #2: Yes

3. Have the authors made all data underlying the findings in their manuscript fully available?

Reviewer #1: Yes

Reviewer #2: Yes

4. Is the manuscript presented in an intelligible fashion and written in standard English?

Reviewer #1: Yes

Reviewer #2: Yes

Reviewer #1: First, I would like to congratulate the research team on their work, as it represents a serious and methodologically sound proposal that brings an innovative perspective to the field of reading research. These comments should be understood as an external viewpoint offering suggestions to be considered in the context of a solid piece of work, rather than as negative criticism.

Required Revisions and Recommended Changes

The manuscript addresses a relevant methodological question and is generally well executed; however, several concrete revisions are required to improve clarity, balance, and methodological transparency.

First, the Introduction should be refocused. Sections describing biases in self-reported measures and diary methods are repeated across paragraphs. These should be condensed, with redundant explanations removed, so that the introduction more clearly leads to the study’s main contribution. The Introduction is relatively brief compared to the rest of the manuscript. Considerable emphasis is placed on reading time, but the manuscript does not sufficiently substantiate why reading time is so important. What positive outcomes does it influence? Reading competence? Why are the European Recommendations on Key Competences (which explicitly address reading competence), or frameworks such as those from the OEI or PIRLS, not mentioned? Likewise, why is there no reference to other existing applications, such as TECLEED? It would be valuable to state explicitly that reading time has a positive impact on reading competence, which in turn positively affects the overall teaching–learning process. Greater reading practice is associated with fewer reading difficulties (see, for example, https://doi.org/10.1371/journal.pone.0295606 and https://doi.org/10.24310/isl.20.2.2025.20620).

Second, the rationale for key methodological decisions must be made explicit. The authors should clearly justify (a) the choice of a 14-day data collection period, (b) the exclusion of reading sessions shorter than one minute, and (c) the use of winsorization for extreme values. Each decision should be briefly supported by references, prior literature, or sensitivity considerations to improve reproducibility.

Third, statistical reporting should be strengthened. Effect sizes should be systematically reported alongside p-values for all main comparisons and regression models. In addition, the handling of missing data through grade-level mean imputation requires further justification, as this method may reduce variance and affect correlations with reading fluency. A short discussion of its potential impact should be added.

Fourth, the Results section should be streamlined. Detailed numerical descriptions that do not directly address the research questions—particularly repeated confirmations of non-significant grade-level differences—should be shortened or moved to supplementary materials to improve readability and focus.

Fifth, the Discussion needs to be more balanced. While the advantages of the logged measure are convincingly presented, the limitations of app-based data collection should be stated more explicitly. These include parental compliance, the possibility of unrecorded independent reading, age-related supervision differences, and potential reactivity effects. Addressing these limitations would strengthen the credibility of the conclusions.

Sixth, the implications section should be expanded. The authors should clarify how the findings apply to large-scale, longitudinal, or cross-cultural studies, and discuss practical constraints related to implementing app-based logging in broader samples.

Finally, minor editorial revisions are required. Several sentences are overly long and should be simplified, repeated definitions (e.g., of the logged measure) should be reduced, and clearer subheadings in the Discussion would improve structure and reader guidance.

Overall, the manuscript is strong, but I think these specific revisions are necessary to enhance clarity, methodological rigor, and overall impact.

Reviewer #2: The proposal fits within the journal’s thematic scope and offers an innovative perspective; therefore, it could be considered for publication. Below, I provide my perspective with suggestions for improvement.

The introduction is clear and logically structured, although somewhat limited in scope. For instance, what does dedicating time to reading imply in relation to the development of reading habits? In other words, the importance of reading time is assumed, but the manuscript does not explain with sufficient clarity why it is a relevant construct in educational, social, economic, or other terms. A revision is recommended to consolidate repeated concepts (e.g., biases) and improve cohesion. Does reading time influence reading competence (see the distinction between reading comprehension and reading competence)? Does it affect critical competence (https://doi.org/10.24310/isl.vi18.15839)?

It might also be useful to include a brief comparison with other existing applications or tools. Additionally, does reading time affect reading development (https://doi.org/10.1371/journal.pone.0193450)?

International reading frameworks are not mentioned, nor are studies showing that family reading promotion is a decisive positive factor, despite existing evidence supporting this claim. Furthermore, there is a documented relationship between reading and the development of self-concept at this educational stage (https://doi.org/10.24310/isl.19.1.2024.17434), which could strengthen the theoretical framework.

The description of the participants is clear and complies with appropriate

ethical principles. However, the sample size is relatively small (109 boys and

girls), which may limit the generalizability of the findings and therefore requires

stronger justification.

The study describes technically sound scientific research, and the data support the conclusions. However, some questions arise: could it be that parents overestimate reading time because their children merely “appear” to be reading? How was this possibility controlled? Moreover, the key contribution—the use of a logged measure via a stopwatch-based app—appears relatively late in the manuscript. It would be advisable to introduce this contribution earlier in the introduction to better guide the reader.

Finally, could this study be applied in school contexts, reading intervention programs, the monitoring of reading habits, or the evaluation of educational policies?

I recommend revising the citations and references, as well as shortening some overly long sentences.

.

Reviewer #1: No

Reviewer #2: No

---

## [Author Response · Author response to Decision Letter 1]

22 Feb 2026

Dear Editor,

We sincerely appreciate the opportunity to submit a revision of our manuscript. We are very grateful to the two anonymous reviewers for their thorough evaluations and insightful comments.

Below, we provide a detailed account of how we have addressed each of the reviewers’ points. Given that the revisions have resulted in substantial changes in the manuscript, we have highlighted the major modifications made in response to specific comments by marking the text in red. In this letter of response, we also indicate where the revised passages can be found (in the manuscript without track changes).

We hope that these revisions meet your and the reviewers’ expectations. Please do not hesitate to reach out if any further modifications are needed.

Sincerely,

Brice Brossette, on behalf of all authors.

Reviewer 1

1) First, the Introduction should be refocused. Sections describing biases in self-reported measures and diary methods are repeated across paragraphs. These should be condensed, with redundant explanations removed, so that the introduction more clearly leads to the study’s main contribution.

RESPONSE: We thank the reviewer for this helpful suggestion. We have revised the Introduction to reduce redundancy in the description of limitations associated with retrospective self-report questionnaires and reading diary methods. Specifically, overlapping explanations of cognitive, memory-related, and social desirability biases were condensed, and the limitations of diary methods were streamlined to avoid repetition across paragraphs (see p. 4-5).

2) The Introduction is relatively brief compared to the rest of the manuscript. Considerable emphasis is placed on reading time, but the manuscript does not sufficiently substantiate why reading time is so important. What positive outcomes does it influence? Reading competence? Why are the European Recommendations on Key Competences (which explicitly address reading competence), or frameworks such as those from the OEI or PIRLS, not mentioned? Likewise, why is there no reference to other existing applications, such as TECLEED? It would be valuable to state explicitly that reading time has a positive impact on reading competence, which in turn positively affects the overall teaching–learning process. Greater reading practice is associated with fewer reading difficulties (see, for example, https://doi.org/10.1371/journal.pone.0295606 and https://doi.eorg/10.24310/isl.20.2.2025.20620).

RESPONSE: We thank the reviewer for this helpful comment. The Introduction has been expanded to better justify the importance of reading time. We now explicitly situate the study within the framework of public policies promoting literacy, referring to the European Recommendations on Key Competences, the OEI framework, and large-scale assessments such as PIRLS. We have also strengthened the theoretical rationale linking reading time—and more broadly print exposure—to reading competence. The revised text explicitly states that greater reading practice is associated with higher reading competence and fewer reading difficulties, supported by the suggested references. In addition, we now discuss the reciprocal nature of this relationship (i.e., the “snowball effect”), whereby better readers read more, and increased reading further enhances competence (see p. 3-4).

Regarding existing tools, we clarify that, to our knowledge, apart from the ambulatory assessment study by Locher et al. (2023), comparable logged measures specifically designed to quantify children’s reading time in naturalistic contexts remain scarce. We carefully searched for references related to the tool TECLEED mentioned by the reviewer; however, we were unable to access scientific publications or detailed methodological descriptions that would allow for an informed comparison.

3) Second, the rationale for key methodological decisions must be made explicit. The authors should clearly justify (a) the choice of a 14-day data collection period, (b) the exclusion of reading sessions shorter than one minute, and (c) the use of winsorization for extreme values. Each decision should be briefly supported by references, prior literature, or sensitivity considerations to improve reproducibility.

RESPONSE: We thank the reviewer for this suggestion. The rationale for the 14-day data collection period (a) and for the use of winsorization for extreme values (c) has now been explicitly added to the main text, together with supporting references (see p. 8 / p. 10). The rationale for excluding reading sessions shorter than one minute (b) has been clarified in a footnote (see p. 10).

4) Third, statistical reporting should be strengthened. Effect sizes should be systematically reported alongside p-values for all main comparisons and regression models.

RESPONSE: We thank the reviewer for this helpful suggestion. In response, we have strengthened statistical reporting throughout the Results section by systematically reporting effect sizes alongside p-values for all main analyses. Specifically, for the comparison between logged and self-reported reading time, we now report Cohen’s d (paired) in addition to the t statistic and p-value. For all regression models, we now report both unstandardized regression coefficients (b) and standardized coefficients (β), allowing interpretation in the original units as well as comparison of effect magnitudes across predictors. In addition, partial R² values are reported for each predictor, quantifying the unique variance explained by each variable beyond the other covariates in the model. These additions provide a more complete and transparent quantification of the magnitude and practical significance of the reported effects (see p .10-14).

5) In addition, the handling of missing data through grade-level mean imputation requires further justification, as this method may reduce variance and affect correlations with reading fluency. A short discussion of its potential impact should be added.

RESPONSE: We acknowledge that mean imputation may attenuate correlations involving the self-reported measure and could therefore marginally increase the observed difference between associations. However, because the proportion of missing data was very small, we believe this effect is limited and unlikely to drive the main finding that the logged measure shows stronger predictive validity. The manuscript has been revised accordingly (see p. 18-19).

6) Fourth, the Results section should be streamlined. Detailed numerical descriptions that do not directly address the research questions—particularly repeated confirmations of non-significant grade-level differences—should be shortened or moved to supplementary materials to improve readability and focus.

RESPONSE: We thank the reviewer for this helpful suggestion. In the revised manuscript (p. 10-14), the Results section has been streamlined to focus more directly on the research questions. Detailed numerical descriptions of grade-level comparisons have been reduced, and grade effects are now summarized with a single statement indicating the absence of significant differences across grades. We chose to retain Table 1 in the main text for descriptive and transparency purposes. Although grade-level differences are not central to our aims, the table provides an overview of the distribution and variability of reading time across grades for both measurement methods, which may be informative for readers interested in developmental patterns or sample comparability. These changes improve readability while preserving essential descriptive information and maintaining a clear focus on the comparison between logged and self-reported measures and their predictive validity.

7) Fifth, the Discussion needs to be more balanced. While the advantages of the logged measure are convincingly presented, the limitations of app-based data collection should be stated more explicitly. These include parental compliance, the possibility of unrecorded independent reading, age-related supervision differences, and potential reactivity effects. Addressing these limitations would strengthen the credibility of the conclusions.

RESPONSE: We thank the reviewer for this comment. The Discussion has been restructured and now includes a specific subsection dedicated to the limitations of the logged measure. This section explicitly addresses parental compliance, the possibility of unrecorded independent reading, age-related differences in supervision, and potential reactivity effects associated with app-based data collection (see p. 15-16).

8) Sixth, the implications section should be expanded. The authors should clarify how the findings apply to large-scale, longitudinal, or cross-cultural studies, and discuss practical constraints related to implementing app-based logging in broader samples.

RESPONSE: We thank the reviewer for this suggestion. The Discussion has been restructured and now includes a new subsection specifically dedicated to the implications and challenges of using logged measures. In this section, we clarify how the findings may extend to large-scale, longitudinal, and cross-cultural research. We also explicitly discuss practical constraints related to implementing app-based logging in broader and more diverse samples (e.g., compliance, scalability, access, and sustainability). In addition, we outline potential methodological improvements and suggest new applications of logged measures in educational and research contexts (see p. 20-21)

9) Finally, minor editorial revisions are required. Several sentences are overly long and should be simplified, repeated definitions (e.g., of the logged measure) should be reduced, and clearer subheadings in the Discussion would improve structure and reader guidance.

RESPONSE: We thank the reviewer for this remark. The manuscript has been carefully edited to improve clarity and conciseness. Clearer subheadings have been introduced in the Discussion to improve structure and reader guidance.

Reviewer 2

10) The introduction is clear and logically structured, although somewhat limited in scope. For instance, what does dedicating time to reading imply in relation to the development of reading habits? In other words, the importance of reading time is assumed, but the manuscript does not explain with sufficient clarity why it is a relevant construct in educational, social, economic, or other terms. A revision is recommended to consolidate repeated concepts (e.g., biases) and improve cohesion. Does reading time influence reading competence (see the distinction between reading comprehension and reading competence)? Does it affect critical competence (https://doi.org/10.24310/isl.vi18.15839)? It might also be useful to include a brief comparison with other existing applications or tools. Additionally, does reading time affect reading development (https://doi.org/10.1371/journal.pone.0193450)? International reading frameworks are not mentioned, nor are studies showing that family reading promotion is a decisive positive factor, despite existing evidence supporting this claim. Furthermore, there is a documented relationship between reading and the development of self-concept at this educational stage (https://doi.org/10.24310/isl.19.1.2024.17434), which could strengthen the theoretical framework.

RESPONSE: We thank the reviewer for this detailed and constructive comment. The Introduction has been substantially enriched to broaden its scope and clarify the educational relevance of reading time (see p. 3-4). First, we now situate the study within the context of public policies promoting literacy, explicitly referring to the European Recommendations on Key Competences (ERKC), the OEI framework, and large-scale international assessments such as PIRLS. This clarifies the central role of reading competence in educational, social, and economic contexts. Second, we have strengthened the theoretical framework linking reading time—and more broadly, print exposure—to reading development. The revised Introduction explicitly states that greater reading practice supports the development of reading skills, and that this relationship is reciprocal. We now discuss the cumulative “snowball effect,” whereby better readers read more, and increased reading further enhances competence. We also highlight its motivational implications, particularly its role in fostering positive academic self-concept, which is crucial at this educational stage. The role of family promotion of print exposure as a key positive factor is now explicitly addressed and supported with relevant references.

Regarding existing tools, we clarify that, to our knowledge, apart from the ambulatory assessment study by Locher et al. (2023), comparable logged measures of reading time are scarce. While educational applications exist to promote reading learning, they fall outside the scope of the present study, which focuses specifically on measurement rather than intervention.

11) The description of the participants is clear and complies with appropriate ethical principles. However, the sample size is relatively small (109 boys and girls), which may limit the generalizability of the findings and therefore requires stronger justification.

RESPONSE: We thank the reviewer for this important comment. To better justify the sample size, we relied on prior literature to estimate the expected magnitude of the association between reading time and reading fluency.

First, Torppa et al. (2020) provide a relevant benchmark for estimating the expected effect size. In a large longitudinal sample of children from Grade 1 to Grade 4, the authors reported correlations between reading fluency and leisure reading ranging from r = .28 to r = .41 (mean r ≈ .35), based on self-reported reading behavior.

These results are consistent with the meta-analysis by Mol and Bus (2011), which shows that associations between print exposure and reading fluency (or word-level reading outcomes) are generally stronger than those observed for more constrained basic decoding skills.

Adopting a conservative approach, we therefore assumed an expected correlation of r = .28 and a conventional target power of 80%, in line with standard recommendations in the behavioral sciences (Cohen, 1992). An a priori power analysis conducted using the pwr.r.test function from the pwr package in R indicated that a sample size of 97 participants would be sufficient to detect such an effect (two-sided α = .05).

In addition, if one computes the required sample size based on the observed correlation between self-reported reading time and fluency (r = .25), the corresponding estimate is 123 participants, which is relatively close to the current sample size (N = 109). Finally, a post hoc power estimate based on the observed effect size (r = .25) and the actual sample size indicates a statistical power of approximately 75%.

Taken together, these analyses suggest that the present study was reasonably powered to detect theoretically and empirically plausible associations between reading time and reading fluency. The manuscript has been revised accordingly, and an explanatory footnote has been added (see p. 10-14).

12) The study describes technically sound scientific research, and the data support the conclusions. However, some questions arise: could it be that parents overestimate reading time because their children merely “appear” to be reading? How was this possibility controlled? Moreover, the key contribution—the use of a logged measure via a stopwatch-based app—appears relatively late in the manuscript. It would be advisable to introduce this contribution earlier in the introduction to better guide the reader.

RESPONSE: First, the issue of children merely “appearing” to read is now explicitly acknowledged in the Discussion. We recognize that the current design does not allow us to directly assess the depth of children’s engagement during reading episodes, and we clarify this as a limitation of the study (see p. 17).

Second, the Introduction has been revised to introduce the key contribution earlier. After presenting the relationship between reading time and reading skills, we now provide a clear roadmap paragraph that explicitly highlights the contrib

---

## [Editor Report · Decision Letter 1]

27 Feb 2026

Measuring Reading Time: Comparing Logged and Self-Reported Data in Relation to Reading Skills

PONE-D-25-58106R1

Dear Dr. Brossette,

We’re pleased to inform you that your manuscript has been judged scientifically suitable for publication and will be formally accepted for publication once it meets all outstanding technical requirements.

Kind regards,

Elena del Pilar Jiménez-Pérez, Ph.D.

Academic Editor

PLOS One
---

## [Editor Report · Acceptance letter]

PONE-D-25-58106R1

PLOS One

Dear Dr. Brossette,

I'm pleased to inform you that your manuscript has been deemed suitable for publication in PLOS One. Congratulations! Your manuscript is now being handed over to our production team.

Kind regards,

on behalf of

Dr. Elena del Pilar Jiménez-Pérez

Academic Editor

PLOS One